# Old Mine Map Georeferencing: Case of Marsigli's 1696 Map of the Smolník Mines

Ladislav Hvizdák [1],*, Dana Tometzová [1], Barbora Iannaccone [1], Marieta Šoltésová [1], Lucia Domaracká [2] and Kamil Kyšeľa [3]

1. Department of Geotourism and Mining Tourism, Institute of Earth Resources, Faculty of Mining, Ecology, Process Control and Geotechnology, Technical University of Košice, Letná 9, 042 00 Košice, Slovakia; dana.tometzova@tuke.sk (D.T.); barbora.iannaccone@tuke.sk (B.I.); marieta.soltesova@tuke.sk (M.Š.)
2. Department of Earth Resources Management, Institute of Earth Resources, Faculty of Mining, Ecology, Process Control and Geotechnology, Technical University of Košice, Letná 9, 042 00 Košice, Slovakia; lucia.domaracka@tuke.sk
3. Institute of Geodesy, Cartography and Geographical Information Systems, Faculty of Mining, Ecology, Process Control and Geotechnology, Technical University of Košice, Letná 9, 042 00 Košice, Slovakia; kamil.kysela@tuke.sk
* Correspondence: ladislav.hvizdak@tuke.sk

**Abstract:** Historical maps represent a unique and irreplaceable source of information about the history of a country, be it large (historical) regions, individual geomorphological units or specifically defined sites. Using a methodologically correct, critical historical analysis, old maps provide both the horizontal and vertical analysis of a landscape and its transformation in different time periods. These maps represent some of the oldest, but relatively easily accessible, historical pictorial documents (plausibly) depicting historical landscapes. This study provides the methodology for processing and georeferencing old mine maps with the possibility of their further use for the purposes of mining tourism. The 1696 Marsigli mine map has been chosen for the case study in question. It depicts a cross-section of the copper mines in Smolník and shows in detail the process of cementation water mining. Through an analysis and a detailed study, two-dimensional parts of a georeferenced historical map have been plotted in Google Earth's three-dimensional space.

**Keywords:** georeferencing; historical mine map; Marsigli; cementation; Smolník; Schmelnitz

## 1. Introduction

Historical maps represent one of the most important materials of great capacity to inform, which can, at the same time, be used for the needs of several scientific disciplines, including land use science and landscape ecology [1]. A scientific work published by the University of Chicago Press [2–6] looks at the issue of old maps in more detail. The greatest advantage of historical maps is that they capture the phenomenon under review in a time–space context [7–10]. The authors of "History of Cartography" consider maps (as well as books) to be "agents of historical changes" [2–6]. Historical maps kept in digital archives contain not only planimetric, but also digitalisation process-related errors [11]. When performing exploration and prospecting work, opening deposits and, last but not least, mining, an important prerequisite was the thorough knowledge of topography of deposits and mine works. This gradually led to the creation of the first mine maps. Mine maps are very special cartographic works. Krokusová, whose research deals with mine maps, argues that this kind of map is different from others, both technically and cartographically, and may initially appear to be incomprehensible [12]. There was no standardisation of cardinal directions introduced into the earliest maps. At that time, it was not common to use maps, and the maps that were used usually depicted only small parts of a landscape. The conventional elements they contained were only simple graphic drawing elements [13,14].

The first mentions of drawing ore veins and mines into the maps of the territory of Slovakia are from the 16th century [15]. The knowledge of minerals, ores, ore veins as well as mining activities gradually determined the need for schematic drawings and mine maps [16,17]. The map is thus one of the manifestations of special historical experience and rich traditions, as well as advanced measuring technology associated with the boom of ore mining in Slovakia in the 16th century [18]. The oldest known mine map made in Slovakia is the map of the Boží Dar adit in Jarabá, Low Tatras, which is also the oldest mining map of the Hofkammerarchiv in Vienna, where it can be found [19]. It was an important document of mining and (partly) geological procedures and, at the same time, the most significant 16th century cartographic work prepared in Slovakia [20]. Since the middle of the 17th century, there have been ever-increasing data on the progress of mining works, solutions of mining-related geological issues and other operating conditions that were based on mine maps [21–23]. The oldest large-scale historical maps of the territory of Slovakia were plotted in the first half of the 18th century by mining land surveyors and mainly depicted the surroundings of mining towns (Banská Štiavnica, Kremnica, Banská Bystrica, Ľubietová, Smolník, Gelnica and others). One of those who contributed to the existence of such maps was Samuel Mikovíni, who also created medium-scale maps of the Hungarian counties of the same period. The 18th century map makers were mostly mining experts and cartographers such as S. Mikovíni, Kristián Fridrich Angerstein and M. Zipser. Another mining expert was A. Péch who prepared a proposal for a simplified method of mine map plotting in the 19th century. A renowned expert who contributed to the archaeological, botanical, hydrogeological, zoological and climatological research of the territory of Slovakia was Luigi Ferdinando Marsigli from Bologna [24–27]. A crucial part of his work focusing on Slovakia was dedicated to mining, mineralogy and geology. There, he described key Slovak mining sites, technology, geological conditions and the locations of mineral and fossil deposits. Moreover, he was the author of the first mineralographic map of Slovakia.

## 2. Study Area

For the purpose of this paper, historical maps, data, and case studies of the Smolník mining landscape and its surroundings were used [28–30]. The selected historical mining town, which formerly belonged to the Union of Upper Hungarian Mining Towns, is located in the south-eastern part of Slovakia, in the Košice region (Figure 1).

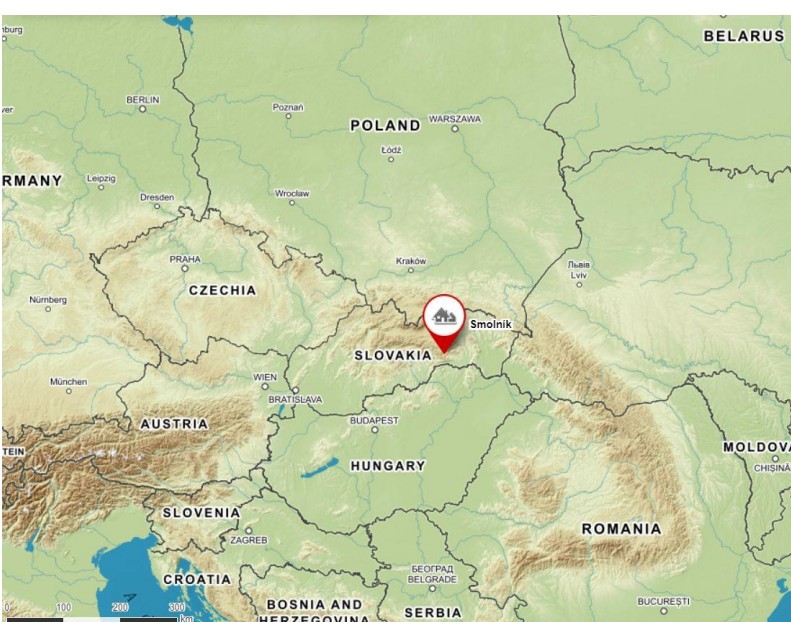

**Figure 1.** Location of the studied area of the mining town Smolník within Slovakia and Europe [31] modified by the author.

The Union of seven mining towns of Upper Hungary was officially founded on 26 December 1478 in Košice and was formed by the towns of Gelnica, Jasov, Rožňava, Smolník, Spišská Nová Ves (today's Slovakia), Rudabánya and Telkibánya (today's Hungary).

## 3. Research Aims and Questions

The main objectives of this study are to georeference the 1696 Marsigli mine map, describe the methodology of georeferencing and present the possible visual use of old mining maps. To meet the objectives, the following questions were asked:

Q1:   Is it possible to georeference and place any mine map in real space?
Q2:   What is the degree of inaccuracy of georeferencing old mining maps and what importance does it play?
Q3:   What is the potential use of georeferenced historical mining maps?

## 4. Materials and Methods

The research methodology can be divided into three stages. The first is the identification of the place and site shown on the map, which may not be clear to start with. The second stage covers both the obtaining of historical and contemporary maps depicting the selected sections and the historical research and study focusing on the author of the maps and maps themselves. An equally important part of this stage is the archival research of the settlement area, in our case the royal mining town of Smolník, and the cementation process. As part of the evaluation process, we tried to work with the most accurate historical source material so as to be able to work with the historical state of the site based on georeferenced historical maps. The third stage focuses on working with maps in the ESRI ArcGIS 10.3 program [32]. The final output is to translate this more than 300-year-old Marsigli's map that shows no geographic coordinates into a real-time map. Using a graphics program, some parts of Marsigli's map have thus been successfully introduced into the Google Earth tool.

### 4.1. Luigi Ferdinando Marsigli

Luigi Ferdinando Marsigli (1658–1730) was an aristocrat, Habsburg general and scientist from Bologna [24,25,27]. He served as a military engineer in the Imperial Army during the battles for the liberation from Turks. In 1691, he was elected a member of the Royal Society of England and was asked to study and collect data on the geography of Hungary [26]. He mapped the 850 km long Habsburg–Ottoman border in the then Kingdom of Hungary, today's Croatia, Serbia and Romania included. In those twenty years that he spent in Hungary researching the river Danube, he gathered scientific information and samples, took measurements and made observations [33]. He described the waterways of the Danube while, at the same time, stating the results of astronomical and meteorological measurements [34]. He was also interested in mineral water springs and baths, especially in their healing benefits. Moreover, Marsigli described, back then, mysterious cementation waters in Špania Dolina and Smolník. He dedicated his work to archaeological research, the study of fish, birds [35], plants, mammals and insects. He published the knowledge he had gained in a book called *Danubius Pannonico-mysicus, observationibus, geographicis, astronomicis, hydrographicis historicis, physicis*, which full title in English is "*Geographic, astronomic, hydrographic, historic, and physical observations on the Mysian Pannonian*". The work comprised six volumes and was published in Hague in 1726 [36]. In the Volume 3, he described the mineral resources of the Danube and its surroundings, as well as mining activities in Hungary and Transylvania. The volume consists of eight parts, the third one of which deals mainly with ore mining in Banská Štiavnica, Špania Dolina and Smolník. Attention is also paid to cementation waters in Smolník. In Volume 6, he published the maps he had corrected (Figures 2–4). These were mainly mine maps of Banská Štiavnica, Špania Dolina and Smolník in the above volume. Nevertheless, it also contained the general maps of the entire Danube region and the maps of the Bratislava and Komárno counties [37,38].

### 4.2. Location Identification—Smolník

The beginnings of mining activities in Smolník date back to the 11th century, but the first written mention of Smolník, in which the first data on gold production was given, is from 1243. In 1327, Smolník was given a royal and mining burgh status by King Henry I of Hungary, who also introduced the mining law of Banská Štiavnica there. The first significant boom of Smolník was noticed in the 14th and 15th centuries, when the flourishing mining industry provided jobs for hundreds of miners, metallurgists, merchants and craftsmen from different towns and villages [39]. From 1529 to 1628, the Smolník mines belonged first to the Thurzo family and later to the Csakys. In 1690, the Smolník mines were bought by the state from the Csaky family [40]. The Smolník copper enterprise was thus successfully managed as a state property for 230 years. The Smolník ore district was historically one of the most significant and richest medieval locations of pyrite and chalcopyrite ore deposits in Slovakia [41–43].

Thanks to intensive copper production in Smolník and the wider Spiš mining region in the above time period, this ore was exported from the Spiš-Gemer region even to places outside the Kingdom of Hungary. At the turn of the 13th and 14th centuries, merchants from the territory of today's eastern Slovakia, primarily from the trading towns of Levoča and Košice, transported copper from Smolník or Spiš directly to Prussia. From there, the Spiš copper was further exported to Flanders. Its usual transport route started in Gdansk and ended in Lübeck.

A characteristic phenomenon in the history of Smolník deposit mining was the extraction of cementation copper which was formed by the oxidation of chalcopyrite in the presence of pyrite and by their transformation into sulphates (mainly blue vitriol) which, thanks to their solubility, entered the mine waters through disturbed deposits [44,45]. After its chemical reaction with iron ore, copper precipitated on the iron ore surface. The cementation process was introduced in the Smolník region in the 15th century. As unique Cu-containing raw material, cementation water became a specific feature of copper mining in the Spiš-Gemer Ore Mountains [46]. The cementation process itself remained a mystery until its explanation by electrochemistry in the 19th century. As it was described by Karpenko [47,48], the process of cementation consists of two chemical reactions, where metallic iron (s) is oxidised and is passed into a solution in the form of hydrated iron cations (aq), i.e., with bound water molecules, where $Fe(s) \rightarrow Fe^{2+}(aq) + 2e-$. At the same time, the two electrons produced in this reaction are consumed in the reduction of copper ions originally present in the solution into the metallic copper, which is precipitated on iron, i.e., $Cu^{2+}(aq) + 2e- \rightarrow Cu(s)$. Summing up both reactions, we obtain the aggregate equation of $Fe(s) + Cu^{2+}(aq) \rightarrow Fe^{2+}(aq) + Cu(s)$. This phenomenon is a result of different electrochemical potential values of the two examined systems called $Fe^{2+}/Fe$ and $Cu^{2+}/Cu$. This means that the metal with the more positive potential value will be precipitated from the solution of its salt on the surface of the metal that has more negative potential [45]. Smolník is the only deposit in Slovakia where mine water was used for the production of cementing copper for almost 735 years. Another artifact (vessel made in 1727) that proves the importance and rarity of the waters of Smolník contains an old inscription which reads: "I was iron, I became copper—I contain silver—I am covered with gold" [49]. Due to the lack of water, it was necessary to build a water management system that would supply and accumulate surface water from greater distances. This water system was the largest in the Spiš-Gemer region and the third largest in Slovakia, after Štiavnica and Kremnica. In the Smolník ore district, water power represented the most important energy for centuries. It is believed that the water wheel was already used in the district in the 15th century. The first waterworks were timber dams. These were followed by the construction of covered square-profile canals made of quarry stone and by water tunnels.

The first drawings of the Smolník water management system were translated into the 1696 Marsigli map (Figure 4). There, water wheels over three mining shafts and two water wheels for driving pumping equipment are marked. To ensure a sufficient amount of water,

the construction of the Uhorná water reservoir began in 1769. The reservoir, which was completed in 1780, was both designed and constructed by Italian builders.

The beginning of the 19th century was also an omen of Smolnik's downfall caused by natural disasters, emigration and dying copper mining industry. The situation began to improve in the seventies of the 18th century when the mining activities mostly focused on the extraction of pyrite. The metallurgical production in Smolník covered non-ferrous metal ores such as copper, silver and antimony, and sulphur (nonmetal).

Identifying a place and a site shown on the map may appear to be a simple and therefore sometimes even unnecessary activity. It is the inconsistency of this step that can lead to mistakes. In his study in which he dealt with L.F. Marsigli's mineralographic maps and the mysteries of the mine map, Antal Andras Deak considers the Smolník cementation map to be the map of Banská Štiavnica [38]. Both are former mining towns with a unique history. Adding to the confusion is that both cities had very similar names in their heydays as Banská Štiavnica was called Schemnitz and Smolník Smelnitz (Figures 2 and 3).

On Marsigli's map, we tried to calculate the scale in a simple logical way. From the graphic program, we determined the distance between Smolník and Banská Štiavnica. We measured the same distance on today's map. We determined the scale based on the ratio 1:X, meaning 1 cm on the map equals X cm in reality. The scale of Marsigli's mineralogic map is approximately 1:576,616. However, determining the exact scale can be accomplished by detailed study.

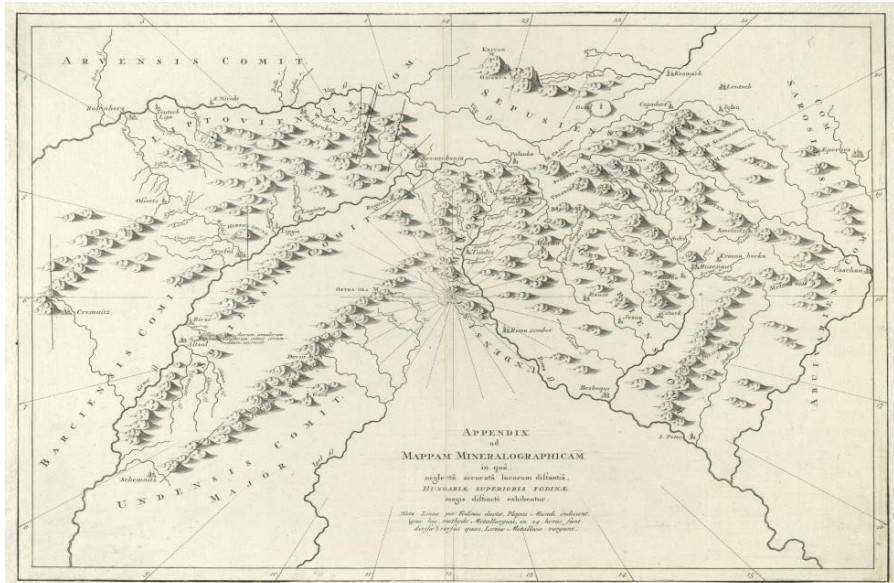

**Figure 2.** Marsigli's mineralographic map of today's Slovakia [50,51].

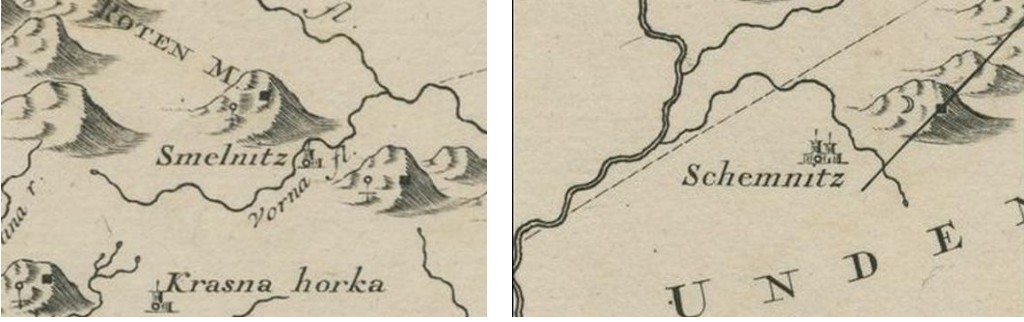

**Figure 3.** Detail from Marsigli's mineralographic map [50,51]. Smelnitz (today's Banská Štiavnica) where black gun powder was used for mining purposes for the first time in 1627. Schemnitz (today's Smolník) known for extracting copper by cementation [50,51].

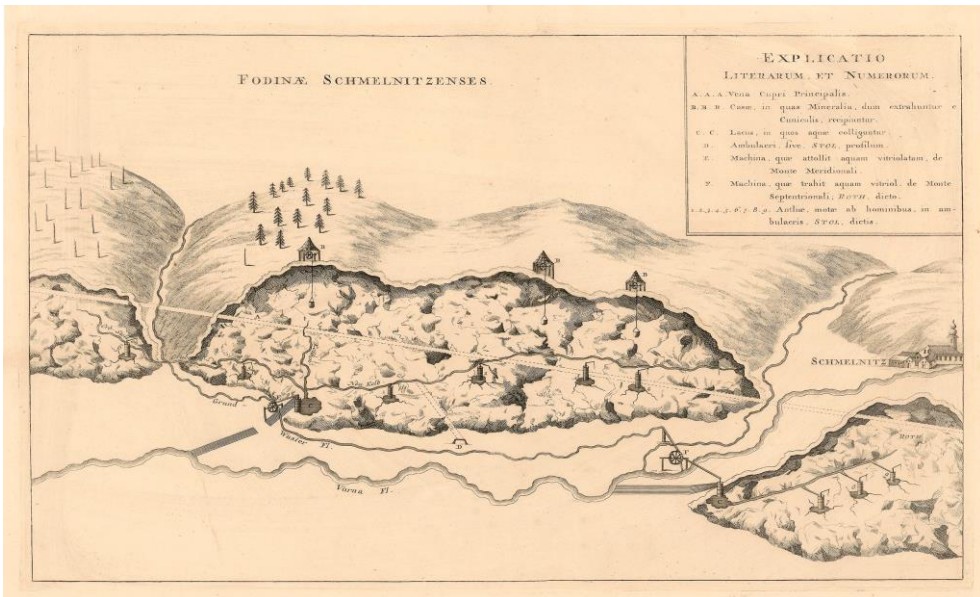

**Figure 4.** Marsigli's map of Fodinae Schmelnitzenses with explanations from 1696. Map orientation: south is at the top of the map [52].

In the 14th century, Smolník received the status of a royal mining town, and the local mining enterprise was one of the most profitable in Hungary. It housed the mining inspectorate, mining court, royal mint and mining school. However, the general decrease in exploration activity and the decline in population meant that it lost its town status and became a village.

*4.3. Description of Marsigli's 1696 Map*

According to Krokusová [12], the period starting from the second half of the 16th century, i.e., from the time when the oldest mining maps were elaborated, to about the middle of the 18th century, represents the first stage in the development of mining maps in Slovakia. A common feature of this period is the way of drawing mine works, especially corridors and galleries. It was the time when simple lines were used to draw mine maps and, especially in the 17th century, holes were punctured in them. From the beginning of the 18th century, levels started to be colour-coded. Amongst other things, excavation activities, horizontal and vertical traffic, and the number of workers at the face began to be graphically depicted on mining works from the second half of the 18th century. However, inclination, geological conditions, direction and ore veins were not yet shown on the maps. If the author of a map wanted to emphasize or mark something, they wrote it in a relevant place. The so-called accompanying text is very necessary when reading the first period maps, as we would not be able to read them without it. In this period, simple and practical mining maps were made, without ornaments or embellishments. Exceptions to this rule were the decorative plant motifs that had their own specific function; namely, to indicate the environment in which the mine work was located. These started to appear on maps in the middle of the 17th century. Since the end of this period, the compass rose, if on the map, was extensively decorated [22]. A fundamental feature of this period in mine maps is the simplicity and practicability of the depiction of corridors, tunnels and other mine elements [53].

Count Luigi Ferdinand Marsigli, a member of the Royal Society in Paris, London and Montpelier, visited Smolník in 1696. In his six-volume work, which was published in Amsterdam in 1726, to be precise, in Volume 3, he also mentioned Smolník cementation waters. In the work above, he also published a map of Smolník and, in Appendix XII, a cross-section of the local mines (Fodinae Schmelnitzences) can be found. The cross-section shows only the surface and the mine, while the ground plan is missing. L.F. Marsigli

admitted that the Smolnik cross-section would require more accurate measurements but added that the arrangement of the depicted sections and the position of individual places, as well as the ore veins, were drawn according to the state existing back then. The map, published by L.F. Marsigli under the name Fodinae Schmelnitzences (Figure 4), is stored in Moll's collection in the Moravian Land Library in Brno [52].

Marsigli's map has been described in detail in Hronček's work; the cementation process included [45]. The map [52] shows the cross-section of the Smolník copper mines used for the extraction of cementation water located under the surface of Spitzen Berg hill (Ostrý vrch, 818 m above sea level). However, the hill itself is not marked and is placed in the middle part of the map. Apart from the description provided by Hronček (based on the original works of Rybár [46,54], Herček [55–60] and Magula [61] that deal with the history of the mining town of Smolník) it is important to mention that L. F. Marsigli used imaging methods very similar or even identical to Georgius Agricola's methods that he applied in his collection of twelve books about mining and metallurgy (Figure 5) [62]. All these historical details are important to identify the points on the map as best as possible.

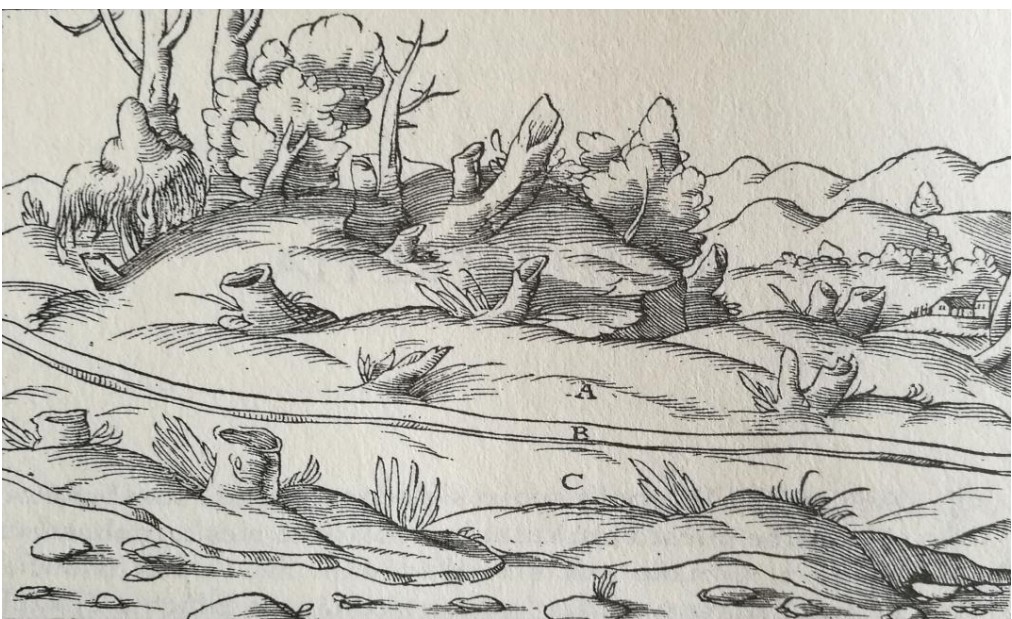

**Figure 5.** Hill A, C. Ore vein B [62].

From the perspective of georeferencing, it is important to find the following information on the map in question:

- Schmelnitz in the name of the map and depicted Smolník mining town;
- Roth in the right bottom section, below Smolník (meaning Rotenberg or Rothenberg, which is the name of the hill used both, back then and today);
- Contemporary names of waterways (brooks and rivers).

The choice of a suitable transformation depends on the quality of the input image, its distortions, which we want to influence in the result, and on the number of identical points. For Masigli's map, we used the affine transformation (1st order polynomial), which geometrically represents the translation, rotation, skew and scaling of each coordinate axis of the original coordinate system [63]. Despite the inaccuracies of the map, Marsigli still thinks that the map shows the actual technical state of cementation. It can be assumed that the lines leading from the shaft to the brook in Smolník (Vorna Fl.) represent a system of six long cementing troughs placed next to each other. Based on the points identified above, it is clear that although the Marsigli's cementation map does not show orientation, it is oriented to the opposite way of today's maps (location of the town of Smolník and the Rotenberg hill).

*4.4. Georeferencing*

Several authors deal with the georeferencing of historical maps, mainly from the 18th century, using different approaches and tools. Hackeloeer, Klasing, Krisp and Meng [64] or Krejčí [65] present a review of methods and applications in georeferencing. Different georeferencing software is compared in a work of Katchadourian and Alberich-Pascual [32]. Worth mentioning are also Molnár [66] or Brovelli and Minghini [67] with their polynomial-based approach, or Cajtlhaml [68] who proposed a new solution on the basis of adjustment of transformation coefficients of all map sheets with the conditions that define the edge continuity. The importance of historical topographic maps as a unique source of information on the state of topography in the 18th century is also supported by specialized geoportals and multimedia websites [69,70]. However, none of the above-mentioned works or websites are dedicated to georeferencing old mine maps. The depiction of mining works from different historical periods on the maps of different quality, scales and geographical coordinates, along with the course of the main veins in the studied area, is made possible by georeferencing (connecting the coordinates of raster files), which is one of the tools of the Geographic Information System (GIS). The system allows maps to be easily rotated or panned, provided that the scanned paper maps have been vectorized into a GIS database. The result is a set of digitized maps depicting the Earth's surface, mine maps from older historical and more recent periods, and the geological structure of a territory under investigation. The digitized maps are then transformed into the Cartesian coordinate system [71]. Another requirement is that the existing artifacts from historical mining activities—gallery portals, pingen (term taken over from the German word "die Pinge" which means mining sinkhole or anthropogenic sinkhole [72,73]) and pingen fields—can be included in the overall picture of mining activities and in the spatial arrangement of ore veins in the monitored area. For the purpose given, it is necessary to identify the location of the artifacts in relation to the exits of ore veins and to coincide the course of historical mining works with the course of ore veins. With the help of georeferencing, the following facts can be displayed in the background map: the course of ore veins, GPS coordinate-based locations of the portals of old galleries which are mostly abandoned and collapsed today, and the location of ore field shafts. To fulfil the goal, it is necessary to select a map that could serve as a background.

The scales of the processed maps can also relate to the different historical periods of their creation. Historical mine and geological maps can be of different scales. The orientation of such historical maps differs from that of the current map documentation where north is depicted on the upper edge of the map. Of course, the scales that seem unusual today are the result of old-time mapping activities. Historical hand-drawn mine maps were characterized by another anomaly compared to today's customs. In connection with some historical mine maps, it should be remembered that each object recorded on such a map can significantly help georeference Marsigli's map. This mainly applies to the maps from earlier time periods that do not use any scale or geographical coordinates, the coordinate systems used nowadays included. In this case, it is necessary to look for a reference point (ground control point = GCP) which, despite the time period, is immutable, such as the floor plan of a church, the portal of a mine gallery, a mine shaft and many other features. Georeferencing the historical map can also be supported by field research. It is necessary to map the monitored territory and record GPS coordinates. Afterwards, these are to be converted into GIS where they can be used as reference points.

In order to obtain historically relevant information from old geographical or mine maps, it is necessary to georeference them in the first step of digitization and subsequent research. This means that the maps have to be positioned in the current coordinate system. Methodologically speaking, this step has to be based on the works by, for example, G. Timár [74], T. Hlásny [75], M. Boltižiar and B. Olah [76–78], J. Cascón-Katchadourian [79], J. Cajthaml [80] and other authors.

Ground control points (GCPs) must be selected in such a way as to minimize the deformation of the background map position, i.e., the points are to be far enough apart that they can be identified with certainty and can evenly cover the area shown on a map. The

number of GCPs depends on the chosen transformation. Of course, the more of them, the higher the assumption of georeferencing accuracy. When georeferencing Marsigli's map, only three ground control points were identified, even though the original intention was to have at least six to eight of them. However, considering the age of the map, its content, the imaging technique used and the preservation of suitable mining relics, the goal could not be reached. The above points were (Figures 6 and 7) the shaft under the Červený vrch hill (Rotenberg), the Kalb gallery and the Jozef shaft at the northern foot of the Ostrý vrch hill (Spitzenberg).

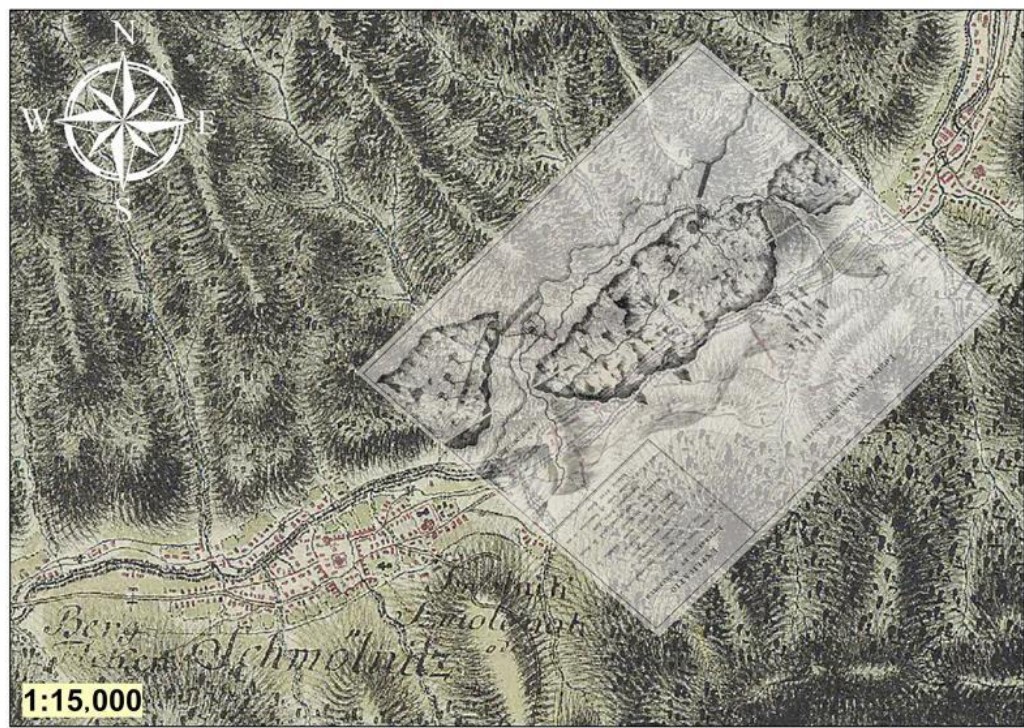

**Figure 6.** Georeferenced Marsigli's map (transparent 40%) with background map of the second military land survey [52,69].

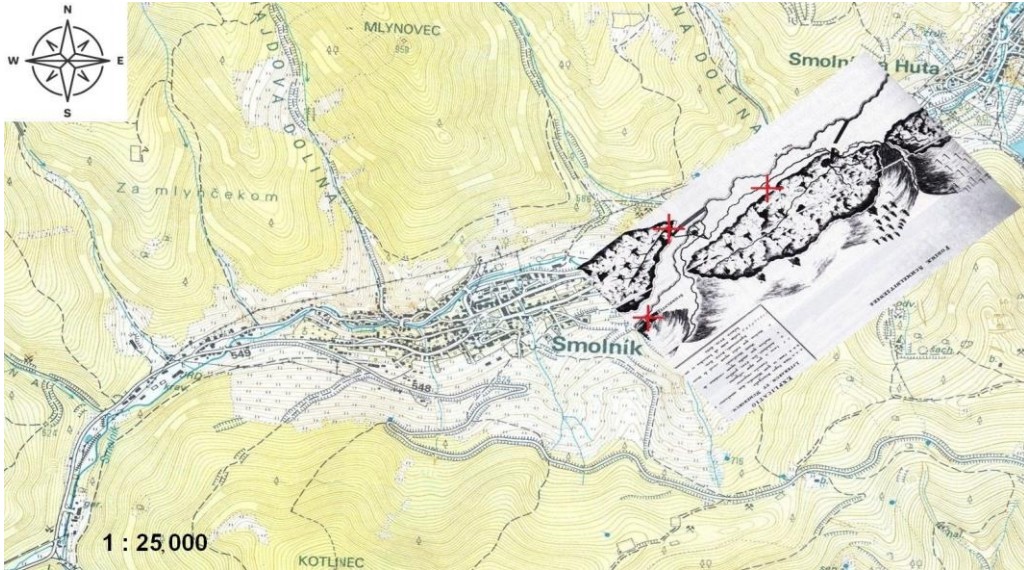

**Figure 7.** Topographic map of Smolník with a georeferenced Marsigli's map.

The ground control points must be identifiable points in the landscape, both in the given time horizon and at present. Their position is determined in a current map or directly on site. Then, they are identified in a historical map and overlapped in the GIS environment (georeferencing) as shown in Figure 6. These are mainly road intersections, key buildings, towers or entrances to mining works.

The modified georeferenced map can then be inserted into a current topographic map and be further used. With the help of professional graphic programs, Adobe Illustrator and Corel Draw, we managed to insert the map processed in this way into the Google Earth 3D environment. The process included cutting out portions from Marsigli's map (specifically the mining area of Rothenberg) and pasting them into the Google Earth environment while preserving the control points.

### 4.5. Errors and Inaccuracies of Georeferencing Old Mine Maps

Old maps show certain inaccuracies attributable to the use of the methods and tools available back then. As these maps are increasingly used to model landscape changes as part of historical research, it is necessary to know the actual errors (deviations from actual values). Their positional accuracy is expressed as the root mean square of all error values identified on a sample of points, which is called the root mean square error (RMSE) and is denoted as $m_{x,y}$ [81]. The stated value must be taken into account when interpreting objects from old maps with regard to the current position of the points.

The following Formula (1) applies when calculating the root mean square error:

$$m_{x,y} = \sqrt{E(\varepsilon^2)} = \sqrt{\frac{\sum \varepsilon_{x,y}^2}{n}} \tag{1}$$

where $E(x)$ represents the mean value of the random variable, $\varepsilon$ represents the error, which is the difference between the coordinates of the checked old map and the actual value (standard) and $n$ is the number of measurements. The value of the root mean square error alone characterizes the entire set of errors. The root mean square error is calculated on the basis of ground control points (GCPs) which are clearly identifiable points that have not changed over time. They can be identified on both the old and current maps (building corners, intersections, railways, etc.). They must be evenly distributed throughout the area of interest. In order to have enough control points, their number is to follow the theory of probability expressed through the central limit theorem [82], based on which a sufficiently large sample (usually $n > 30$) has a distribution similar to the normal distribution. If all the possible errors on an old map had a normal distribution, the theorem would also apply to smaller samples.

Marsigli's map has no geodetic and therefore no projection background, as confirmed in a study by Janata and Cajthaml [83]. The first full-scale mapping (including the territory of today's Slovakia) was ordered by Mary Theresa (1717–1780) in 1763 after the Seven Years' War which the Austrians lost also due to a low quality of maps compared to the Prussian maps. It took 24 years to complete the mapping process. As it was completed during the reign of Joseph II (1741–1790) it is commonly known as the Josephinian Land Survey. Marsigli's map was created before the first military land survey. It is important to mention that the scale used for the map was 1:28,800. However, it was not based on any specific geodetic (mathematical) foundations for which simple measuring tools (compass, tape measures, chains, folding rulers, telescopes, etc.) and the "a la vue" method (observation and estimation by eye) are used. To determine the degree of inaccuracy a larger number of GCPs are needed to be able to use any of the methods or their combinations.

## 5. Results

The result of the authors' efforts is Marsigli's map positioned in real space. Using appropriate editing tools, individual sections were cut out of the original map and inserted into the Google Maps environment for better visualization (Figure 8). The entire map

placement process was implemented using three GCPs. The process also included the study of historical events either related to the author himself or to the area in question. All these data were evaluated, edited and added to the notice boards placed on an educational hiking trail (Figures 9 and 10). One of the indicators of correct map georeferencing in the ArcGIS environment is the total final deformation of a map. A slight deformation can be seen when the map of the second military land survey, conducted between 1764 and 1787, is used as a background. According to the authors, the deformation can be attributed to Marsigli's map and second military land survey inaccuracies. The same procedure was implemented using the current topographical map. It is worth mentioning that the deformation of Marsigli's map is minimal in the case of an accurate background map. It is surprising that various, roughly 320 years-old mapping techniques were so accurate.

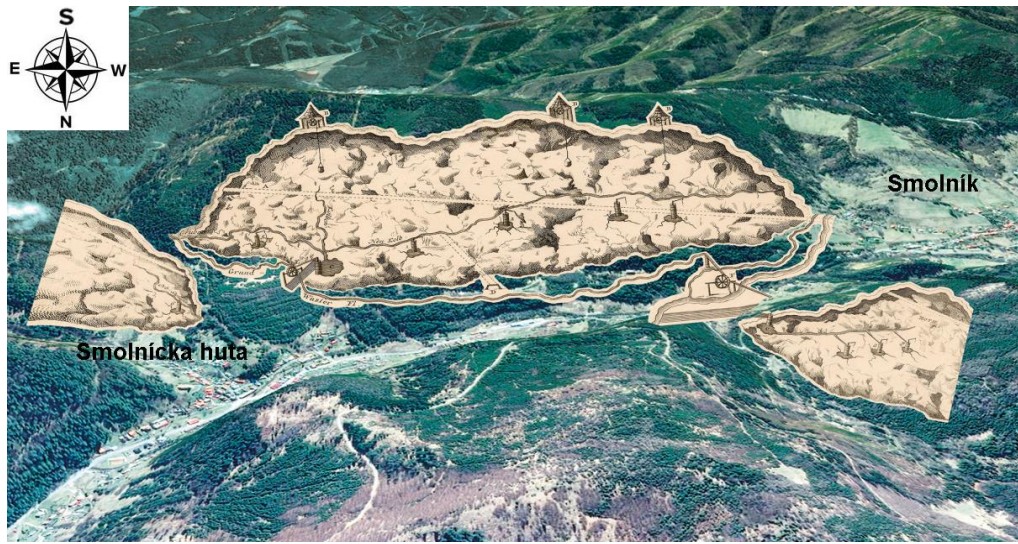

**Figure 8.** Graphical modification of the inserted part of the Marsigli map into the current 3D territory displayed via Google Earth.

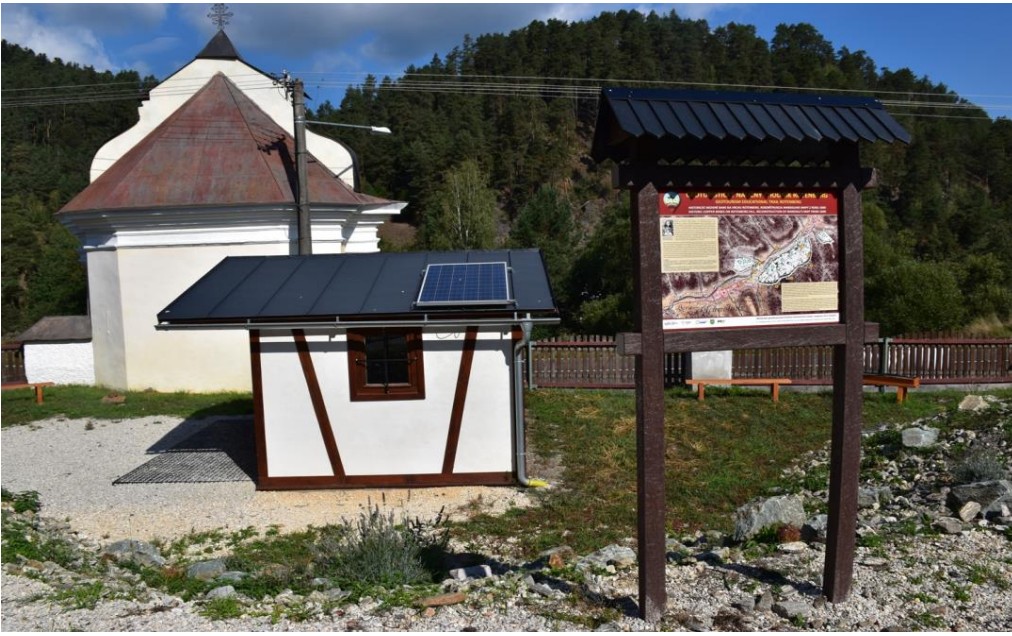

**Figure 9.** Notice board depicting georeferenced Marsigli's map. Rotenberg Geotourist Nature Trail in Smolník [84].

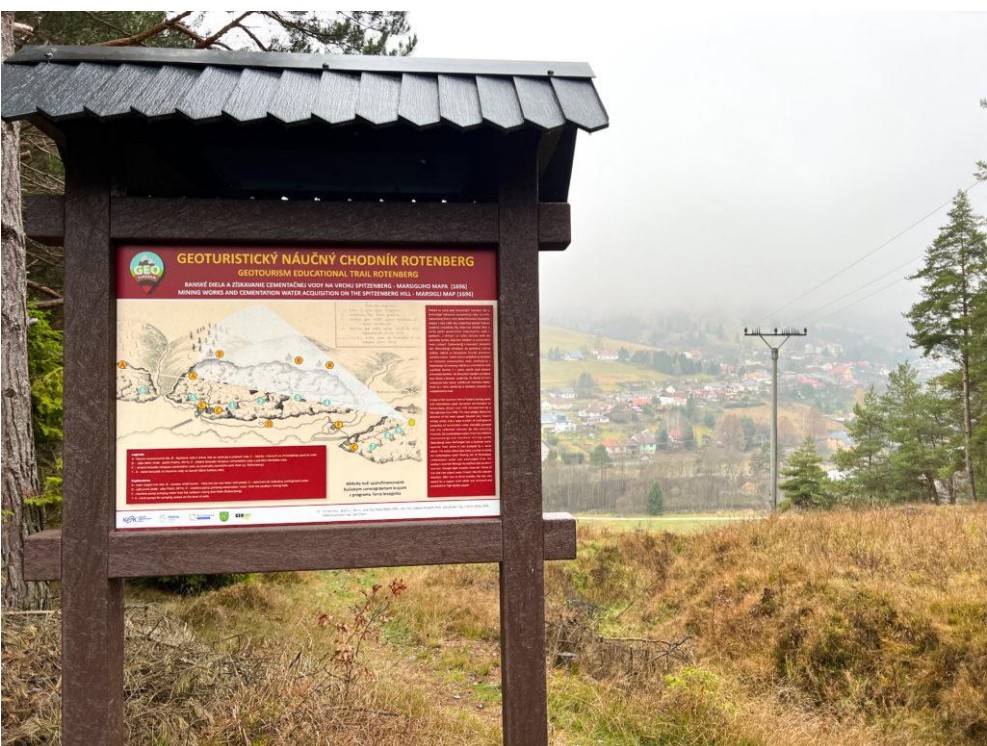

**Figure 10.** Notice board depicting Marsigli's map. Rotenberg Geotourist Nature Trail in Smolník [85].

## 6. Discussion

The proposed research procedure was established for the purpose of mining tourism. The created 2D model (Figure 4) makes it possible to see the basic location of the underground spaces for cementation water extraction in the cross-section of the actual relief of the country as laid out at the end of the 17th century. However, from the perspective of historical mining research it is possible to reconstruct and create a real, relatively accurate picture of the distribution of underground tunnels, the storage of underground technological equipment and the location of surface cementation equipment after proper critical evaluation of the model as seen above. The correctness of these newly created outputs is confirmed by later archived written documents.

Q1: Using this methodological procedure, any mine map can be placed in space. However, the goal will not be achieved if the conditions of the first and second stages mentioned in the introductory parts of the article are not successfully met.

Q2: The first part of the question is answered in the "Errors and Inaccuracies of Georeferencing Old Mine Maps" chapter. The dependence of accuracy of the entire procedure is directly proportional to the purpose for which this map is to be used. Different accuracy is required when using a map for marketing, promotional and educational purposes of the tourism industry, and different for, for example, geodetic purposes. Moreover, the above directly relates to the applied measuring technique. As part of our fieldwork, we used a standard GPS device able to reach an accuracy of a few meters and took into account the work performed in the forest (reduced quality of our GPS signal). In our research, we used GPS in the terrain only for the verification of control points. During this verification, two errors need to be taken into account. The first one is the inherent inaccuracy of GPS devices in the field (we used a standard tourist GPS which, in dense forest cover, sometimes achieves accuracy of less than 5 m, which is sufficient for our purposes). The second error is the inaccuracy that arises during the transformation of GPS coordinates into JTSK coordinates (GIS) using the transformation formulas themselves. Again, for our purposes, this level of accuracy is acceptable.

Q3:　One cannot fully understand the significance of mine maps for historical science and its related disciplines, mining production, and mining and geological exploration. Despite the above importance, these maps remain a completely unused source of information. Mine maps can also tell a lot about mining folklore, art, historical topography, metallurgical and mining production, etc. even though the primary use of mine maps is connected with the history of mining. For centuries their importance for geological and mining exploration has been enormous and they have also played their role in mining production. Nowadays, historical mine maps are widely used due to the fact that we really know which part of the territory they depict in time. Moreover, it is also worth mentioning their educational, marketing, research and publicity purposes (Figures 9 and 10). A study dealing with an appropriate transformation of artistic graphics from historical maps into the third dimension in relation to the creation of perspective landscape images opens up maps to their new potential function [86].

## 7. Conclusions

The use of a 16th century map is a very valuable tool for understanding the spatial dynamics of the Smolník mining town. Recently, there has been an increase in historical source research on landscape development both in our country and abroad, and it has been a widely discussed topic [87,88]. Thanks to their informative value, historical maps offer the possibility of applying several methodological procedures and searching for new approaches. Until now, the comprehensive approach has only been applied to the interpretation of historical map documents to a small extent. When researching the history of mining and related technological processes (the excavation of underground spaces, transport, pumping of mine water, construction of technical equipment, etc.), archival research combined with the critical analysis of written and pictorial historical sources represent a basic methodological procedure. On the basis of these documents, an expert in history, or to be more precise, an expert in the history of mining and technical sciences, can reconstruct the history of the investigated historical, currently non-existent, and often forgotten mining phenomena. These documents are necessary for upgrading historical research, which is currently represented by computer modelling. When modelling and subsequently visualizing the investigated mining phenomena, it is necessary to digitize historically correct documents, as only these will guarantee correct output results. The 2D and 3D outputs created by the presented methodological procedures bring new, very accurate, and to this day unknown historical information about objects, phenomena, technological processes or various structures. To have quality input information for modelling in a computer environment is a fundamental and determining factor for the creation of new relevant and unique historical information. Due to the imperfection of historical data and due to certain doubts related to stated historical facts, we are not always able to make very detailed realistic images. This encourages us to explore the potential of projecting virtual historical landscapes.

**Author Contributions:** Conceptualization, Ladislav Hvizdák and Dana Tometzová; methodology, Ladislav Hvizdák, Dana Tometzová and Barbora Iannaccone; formal analysis, Kamil Kyšeľa; investigation, Ladislav Hvizdák, Barbora Iannaccone and Marieta Šoltésová; resources, Ladislav Hvizdák, Barbora Iannaccone and Marieta Šoltésová; data curation, Marieta Šoltésová and Lucia Domaracká; writing—original draft preparation, Ladislav Hvizdák and Dana Tometzová; writing—review and editing, Dana Tometzová; visualization, Lucia Domaracká and Kamil Kyšeľa. All authors have read and agreed to the published version of the manuscript.

**Funding:** This paper is the result of the project "Research of alternative energy sources' implementation impact on industries of energy management processes" supported by Operational Program Integrated Infrastructure (ITMS: 3131011T564).

**Data Availability Statement:** Not applicable.

**Acknowledgments:** We are thankful for the comments from three anonymous reviewers who helped improve the quality of this paper significantly.

**Conflicts of Interest:** The authors declare no conflict of interest.

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
