# Peer review of "Old Mine Map Georeferencing: Case of Marsigli’s 1696 Map of the Smolník Mines"

_ijgi, doi:10.3390/ijgi12080345_

Round 1

Reviewer 1 Report

Mines of Smolnik

This is an interesting paper based on some research done for a mining interpretation panel and I recommend that it should be published. However, there are some issues with the paper that I outline below. 

It would be interesting to know a bit more about the 'cementation process', in particular the chemical formulae for the reactions and how copper was extracted from the precipitate. 

Line 271. Some additional explanations are required for terms such as 'pingen and pingen fields' - what are these please?

Line 331. The explanation of the equation is not in English - please provide a translation. 

Figure 1. The map appears to be taken from a larger map but without attribution of the source map. Please redraw or provide a reference. 

I also think that the paper would benefit from some more refinement of the English language which seems a little ponderous and somewhat repetitive. This could be accomplished partly by creating a clearer structure for the paper. Currently it is difficult to tell whether the paper intends to outline a new methodology, is an exposition of the mining fields of Smolnik or is a description of a tourist project. I think that the greatest value is as an exposition of the history of mining in the area, and would recommend restructuring to reflect that. I hope that this will also lead to a greater focus with the language. 

I will be happy to provide further feedback, if required. 

The English is satisfactory but is rather discursive and lacks focus. I have made suggestions above on how this might be improved. 

Reviewer 2 Report

331 Slovakian sentence left in the text "Pre výpočet strednej kvadratickej chyby platí vzťah"

Reviewer 3 Report

In general, there is a lot of historical information (very interesting), and relatively little professional information in the field of GIS and compiling and processing spatial data, a description of the specificity of the activities carried out, and thus references to literature in this field.

The style of writing in some parts is like a lecture - how to do. There is no detailed description of what has been done, details about individual stages, information about the selection of parameters or settings.

Other remarks:

Line 75 - With this type of reconstruction based on archival cartographic materials, it is also worth reaching for other sources of information (text, graphic) for verification and supplementation.

Line 85 - Figure 1. - no information about the background material, its origin (what map is it and where did it come from)

Line 86 - Regarding the questions asked - some of them can be answered without the text presented in the article...

Chapter 4.1 and 4.2 - contain a lot of very interesting historical description, but what is the significance of this for the article - it would be necessary to create arguments and smoothly combine with the main goal and idea of ​​the research, which, I suppose, concerns the possibility of compiling spatial data in the aspect of reconstruction of the former landscape

Figure 2.  - when attaching maps, it is worth giving the scale of the original (if it cannot be clearly converted into modern units, give at least an approximate value of the scale)

Figure 4. - very interesting maps – it is worth giving more details at least in the list at the end (e.g.: scale, physical size of the sheet, orientation, authors, storage location with the exact signature…)

Line 246 - jump to georeference - no link to previous content - no text continuity

Line 252 - This information is sufficient to obtain 3 control points, which is the minimum number necessary for georeferencing „ - ... (where does this statement come from, according to whom and what maps does it refer to...?) - it should be developed and explained with reference to the literature; it should also be considered and explained whether the above statements are adequate to the maps used here; according to the reviewer, for non-cartometric studies, 3 control points are not enough.

Line 254 - Despite the inaccuracies of the map…” - it should be expanded on what error values are meant

Line 317 - With the help of professional graphic programs…” - what programs are these and what are they used for? Describe the course of action.

Figure 6 - transparency could be made for the superimposed image, so that the reader could see the relationship/correlation of these two images of space

Line 360 - „The entire map placement process was implemented using a minimal number of GCPs.” - that's not enough GCP points; In addition, the whole process should be described in detail, especially in terms of the selection of transformation parameters, image resolution, etc...

Line 384 - the authors do not compare the results with other studies

Line 403 - the issue of using GPS - there was nothing about it in the text (no details ...)

Line 450 - a large share of literature from the region; the literature resource could be enriched with other valuable publications.

Round 2

Reviewer 2 Report

It is an interesting article on the history of the mining profession

Reviewer 3 Report

Comments in the file.
